# Constrained Reinforcement Learning for Safety-Critical Tasks via Scenario-Based Programming

## Abstract

Deep reinforcement learning (DRL) has achieved groundbreaking successes in various applications, including robotics. A natural consequence is the adoption of this paradigm for safety-critical tasks, where human safety and expensive hardware can be involved. In this context, it is crucial to optimize the performance of DRL-based agents while providing guarantees about their behavior. This paper presents a novel technique for incorporating domain-expert knowledge into a *constrained DRL* training loop. Our technique exploits the *scenario-based programming* paradigm, designed to specify such knowledge in a simple and intuitive way. While our approach can be considered general purpose, we validated our method by performing experiments on a synthetic set of benchmark environments, and the popular robotic mapless navigation problem, in simulation and on the actual platform. Our results demonstrate that using our approach to leverage expert knowledge dramatically improves the safety and performance of the agent.

## 1 Introduction

In recent years, *deep neural networks* (DNNs) have achieved state-of-the-art results in a large variety of tasks, including image recognition (Du, 2018), game playing (Mnih et al., 2013), protein folding (Jumper et al., 2021), and more. In particular, *deep reinforcement learning* (DRL) (Sutton & Barto, 2018) has emerged as a popular paradigm for training DNNs that perform complex tasks through continuous interaction with their environment. Indeed, DRL models have proven remarkably useful in robotic control tasks, such as navigation (Kulhánek et al., 2019) and manipulation (Nguyen & La, 2019; Corsi et al., 2021), where they often outperform classical algorithms (Zhu & Zhang, 2021). The success of DRL-based systems has naturally led to their integration as control policies in safety-critical tasks, such as autonomous driving (Sallab et al., 2017), surgical assistance (Pore et al., 2021), flight control (Koch et al., 2019), and more. Consequently, the learning community has been seeking to create DRL-based controllers that simultaneously demonstrate high *performance* and high *reliability*; i.e., are able to perform their primary tasks while adhering to some prescribed properties, such as safety and robustness.

An emerging family of approaches for achieving these two goals, known as *constrained DRL* (Achiam et al., 2017), attempts to simultaneously optimize two functions: the *reward*, which encodes the main objective of the task; and the *cost*, which represents the safety constraints. Current state-of-the-art algorithms include IPO (Liu et al., 2020), SOS (Marchesini et al., 2021b), CPO (Achiam et al., 2017), and Lagrangian approaches (Ray et al., 2019). Despite their success in some applications, these methods generally suffer from significant setbacks: (i) there is no uniform and human-readable way of defining the required safety constraints; (ii) it is unclear how to encode these constraints as a signal for the training algorithm; and (iii) there is no clear method for balancing cost and reward during training, and thus there is a risk of producing sub-optimal policies.

In this paper, we present a novel approach for addressing these challenges, by enabling users to encode constraints into the DRL training loop in a simple yet powerful way. Our approach generates policies that strictly adhere to these user-defined constraints without compromising performance. We achieve this by extending and integrating two approaches: the *Lagrangian-PPO* algorithm (Ray et al., 2019) for DRL training, and the *scenario-based programming* (SBP) (Damm & Harel, 2001;

Harel et al., 2012b) framework for encoding user-defined constraints. Scenario-based programming is a software engineering paradigm intended to allow engineers to create a complex system in a way that is aligned with how humans perceive that system. A scenario-based program is comprised of scenarios, each of which describes a single desirable (or undesirable) behavior of the system at hand; and these scenarios are then combined to run simultaneously, in order to produce cohesive system behavior. We show how such scenarios can be used to directly incorporate subject-matter-expert (SME) knowledge into the training process, thus forcing the resulting agent's behavior to abide various safety, efficiency and predictability requirements.

In order to demonstrate the usefulness of our approach to safety-critical tasks, we used it to train a policy for performing *mapless navigation* (Zhang et al., 2017; Tai et al., 2017) for robotics by the Robotis Turtlebot3 platform. While common DRL-training techniques were shown to give rise to high-performance policies for this task (Marchesini & Farinelli, 2020), these policies are often unsafe, inefficient, or unpredictable, thus dramatically limiting their potential deployment in real-world systems (Marchesini et al., 2021a;b). Our experiments demonstrate that, by using our novel approach and injecting subject-matter expert knowledge into the training process, we are able to generate trustworthy policies that are both safe and high performance.

To have a complete assessment of the resulting behaviors, we performed a formal verification analysis, following methods such as with (Katz et al., 2017; Liu et al., 2019), of various predefined safety properties that proved that our approach generates safe agents to deploy in *any* environment.

## 2 BACKGROUND

**Deep Reinforcement Learning.** Deep reinforcement learning (Li, 2017) is a specific paradigm for training deep neural networks (Goodfellow et al., 2016). In DRL, the training objective is to find a *policy* that maximizes the *expected cumulative discounted reward* $R_t = \mathbb{E}\left[\sum_t \gamma^t \cdot r_t\right]$, where $\gamma \in [0, 1]$ is the *discount factor*, a hyperparameter that controls the impact of past decisions on the total expected reward. The *policy*, denoted as $\pi_\theta$, is a probability distribution that depends on the parameters $\theta$ of the DNN, which maps an observed *environment state* $s$ to an *action* $a$. Proximal policy optimization (PPO) is a state-of-the-art DRL algorithm for producing $\pi_\theta$ (Schulman et al., 2017). A key characteristic of PPO is that it limits the gradient step size between two consecutive policy updates during training, to avoid changes that can drastically modify $\pi_\theta$ (Schulman et al., 2015).

In mission-critical tasks, the concept of optimality often goes beyond the maximization of a reward, and also involves "hard" safety constraints that the agent must respect. A *constrained markov decision process* (CMDP) is an alternative framework for sequential decision making, which includes an additional signal: the *cost function*, defined as $C : \mathcal{S} \times \mathcal{A} \rightarrow \mathbb{R}$, whose expected values must remain below a given threshold $d \in \mathbb{R}$. CMDP can support multiple cost functions and their thresholds, denoted by $\{C_k\}$ and $\{d_k\}$, respectively. The set of *valid* policies for a CMDP is defined as:

$$\Pi_{\mathcal{C}} := \{\pi_\theta \in \Pi : \forall k, \ J_{C_k}(\pi_\theta) \leq d_k\} \tag{1}$$

where $J_{C_k}(\pi_\theta)$ is the expected sum of the $k^{th}$ cost function over the trajectory and $d_k$ is the corresponding threshold. Intuitively, the objective is to find a policy function that respects the constraints (i.e., is *valid*) and which also maximizes the expected reward (i.e., is *optimal*). A natural way to encode constraints in a classical optimization problem is by using *Lagrange multipliers*. Specifically, in DRL, a possible approach is to transform the constrained problem into the corresponding dual unconstrained version (Liu et al., 2020; Achiam et al., 2017). The optimization problem can then be encoded as follows:

$$J(\theta) = \min_{\pi_\theta} \max_{\lambda \geq 0} \mathcal{L}(\pi_\theta, \lambda) = \min_{\pi_\theta} \max_{\lambda \geq 0} J_R(\pi_\theta) - \sum_K \lambda_k (J_{C_k}(\pi_\theta) - d_k) \tag{2}$$

Crucially, the optimization of the function $J(\theta)$ can be carried out by applying any *policy gradient* algorithm, a common implementation is based on PPO (Ray et al., 2019).

**Scenario-Based Programming.** Scenario-based programming (SBP) (Damm & Harel, 2001; Harel & Marelly, 2003) is a paradigm designed to facilitate the development of reactive systems, by allowing engineers to program a system in a way that is close to how it is perceived by humans —

---

The supplementary material includes the appendices. The code will be released upon publication.

with a focus on inter-object, system-wide behaviors. In SBP, a system is composed of *scenarios*, each describing a single, desired or undesired behavioral aspect of the system; and these scenarios are then executed in unison as a cohesive system.

An execution of a scenario-based (SB) program is formalized as a discrete sequence of events. At each time-step, the scenarios synchronize with each other to determine the next event to be triggered. Each scenario declares events that it *requests* and events that it *blocks*, corresponding to desirable and undesirable (forbidden) behaviors from its perspective; and also events that it passively *waits-for*. After making these declarations, the scenarios are temporarily suspended, and an *event-selection mechanism* triggers a single event that was requested by at least one scenario and blocked by none. Scenarios that requested or waited for the triggered event wake up, perform local actions, and then synchronize again; and the process is repeated ad infinitum. The resulting execution thus complies with the requirements and constraints of each of the individual scenarios (Harel & Marelly, 2003; Harel et al., 2012b). For a formal definition of SBP, see (Harel et al., 2012b).

Although SBP is implemented in many high-level languages, it is often convenient to think of scenarios as transition systems, where each state corresponds to a synchronization point, and each edge corresponds to an event that could be triggered. Fig. 1 uses that representation to depict a simple SB program that controls the temperature and water-level in a water tank (borrowed from (Harel et al., 2012a)). The scenarios *add hot water* and *add cold water* repeatedly wait for WATER LOW event, and then request three times the event Add HOT or Add COLD, respectively. Since these six events may be triggered in any order by the event selection mechanism, new scenario *stability* is added to keep the water temperature stable, achieved by alternately blocking Add HOT and Add COLD events. The resulting execution trace is shown in the event log.

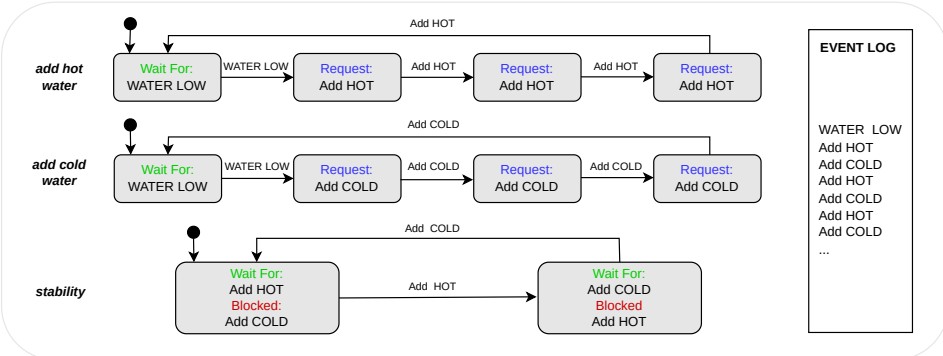

Figure 1: A scenario-based program for controlling a water tank. The small black circle indicates the initial state. *Figure is inspired by the work of Harel et al. (2012a).*

SBP is an attractive choice for the incorporation of domain-specific knowledge into a DRL agent training process, due to being formal, fully executable and support of incremental development (Gordon et al., 2012; Alexandron et al., 2014). Moreover, the language it uses enables domain-specific experts to directly express their requirements specifications as an SB program.

## 3 EXPRESSING DRL CONSTRAINTS USING SCENARIOS

**Mapless Navigation.** We explain and demonstrate our proposed technique using the *mapless navigation* problem, in which a robot is required to reach a given target efficiently while avoiding collision with obstacles. Unlike in classical planning, the robot is not given a map of its surrounding environment and can rely only on local observations — e.g., from lidar sensors or cameras. Thus, a successful agent needs to be able to adjust its strategy dynamically, as it progresses towards its target. Mapless navigation has been studied extensively and is considered difficult to solve. Specifically, the local nature of the problem renders learning a successful policy extremely challenging and hard to solve using classical algorithms (Pfeiffer et al., 2018). Prior work has shown DRL approaches to be among the most successful for tackling this task, often outperforming hand-crafted algorithms (Marchesini & Farinelli, 2020).

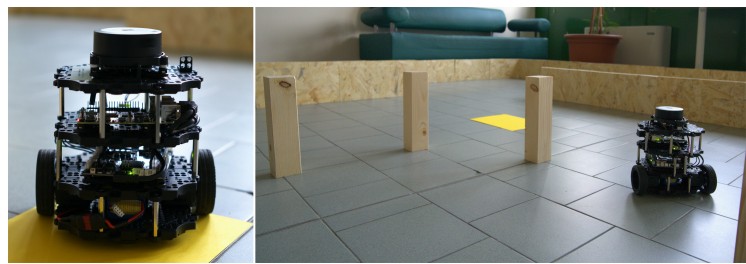

Figure 2: The Robotis Turtlebot3 platform.

As a platform for our study, we used the *Robotis Turtlebot 3* platform (Turtlebot, for short; see Fig. 2), which is widely used in the community (Nandkumar et al., 2021; Amsters & Slaets, 2019). The Turtlebot is capable of horizontal navigation and is equipped with lidar sensors for detecting nearby obstacles. In order to train DRL policies for controlling the Turtlebot, we built a simulator based on the *Unity3D* engine (Juliani et al., 2018), which is compatible with the *Robotic Operating System* (ROS) (Quigley et al., 2009) and allows a fast transfer to the actual platform (*sim-to-real* (Zhao et al., 2020)). We used a hybrid reward function, which includes a discrete component for the terminal states ("collision", or "reached target"), and a continuous component for the non-terminal states. Formally:

$$R_t = \begin{cases} \pm 1 & \text{terminal states} \\ (dist_{t-1} - dist_t) \cdot \eta - \beta & \text{otherwise} \end{cases} \tag{3}$$

Where $dist_k$ is the distance from the target at time $k$; $\eta$ is a normalization factor; and $\beta$ is a penalty, intended to encourage the robot to reach the target quickly (in our experiments, we empirically set $\eta = 3$ and $\beta = 0.001$). Additionally, in terminal states, we increase the reward by 1 if the target is reached, or decrease it by 1 in case of collision. For our DNN topology, we used an architecture that was shown to be successful in a similar setting (Marchesini & Farinelli, 2020): (i) an input layer of nine neurons, including seven for the lidar scans and two for the polar coordinates of the target; (ii) two fully-connected hidden layers of 32 neurons each; and (iii) an output layer of three neurons for the discrete actions (i.e., move FORWARD, turn LEFT, and turn RIGHT). In Section 4, we provide details about the training algorithm we used. Using the reward defined in Eq. 3, we were able to train agents that achieved high performance — i.e., obtained a success rate of approximately 95%, where "success" means that the robot reached its target without colliding into walls or obstacles.

Analyzing the trained agents further, we observed that even DRL agents that achieved a high success rate may demonstrate highly undesirable behavior in different scenarios. One such behavior is a sequence of back-and-forth turns, that causes the robot to waste time and energy. Another undesirable behavior is when the agent makes a lengthy sequence of right turns instead of a much shorter sequence of left turns (or vice versa), wasting time and energy. A third undesirable behavior that we observed is that the agent might decide not to move forward towards a target that is directly ahead, even when the path is clear. Our goal was thus to use our approach to remove these undesirable behaviors.

**A Rule-Based Approach.** Following the approach of (Yerushalmi et al., 2022), we integrated a scenario-based program into the DRL training process, in order to remove the aforementioned undesirable behaviors. More concretely, we created specific scenarios to rule out each of the three aforementioned undesirable behaviors we observed. To accomplish this, we created a mapping between each possible action $a_t \in \{\text{Move FORWARD, Turn LEFT, Turn RIGHT}\}$ of the DRL agent and a dedicated event $e_{a_t} \in \{\text{SBP\_MoveForward, SBP\_TurnLeft, SBP\_TurnRight}\}$ within the scenario-based program. These events allow the various scenarios to keep track and react to the agent's actions. Similarly to (Yerushalmi et al., 2022), we refer to these $e_{a_t}$ events as *external events*, indicating that they can only be triggered when requested from outside the SB program proper. By convention, we assume that after each triggering of a single, external event, the scenario-based program executes a sequence of internal events (a *super-step* (Yerushalmi et al., 2022)), until it returns to a steady-state and then waits for another external event.

The novelty of our approach, compared to (Yerushalmi et al., 2022), is in the strategy by which we use scenarios to affect the training process. Specifically, we define the DRL cost function to

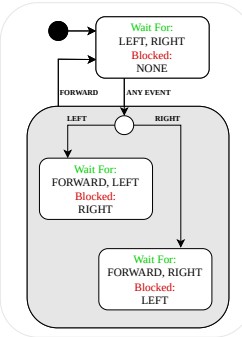 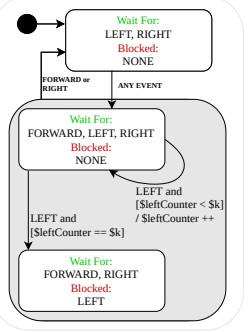 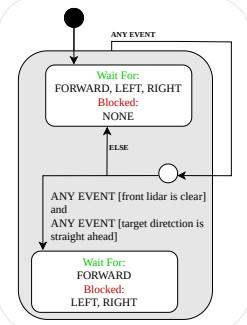

(a) *avoid back-and-forth rotation*  (b) *avoid turns larger than 180°*  (c) *avoid turning when clear*

Figure 3: A visualization of the three scenarios. Figure (b) refers to the *Left turns part* only. 'Wait For' and 'Blocked' in the state-blob indicates events that the scenario waits for or blocks, respectively. The events `SBP_MoveForward`, `SBP_TurnLeft` and `SBP_TurnRight` are represented respectively, by FORWARD, LEFT, RIGHT.

correspond to violations of scenario constraints by the DRL agent. Whenever the agent selects an action that is mapped to a *blocked* SBP event, we increase the *cost*. This approach is described further in Section 4, and constitutes a general method for injecting explicit constraints (expressed, e.g., by scenarios) directly into the policy optimization process.

**Example: Constraint Scenarios.** Considering again our Turtlebot mapless navigation case study, we created scenarios for discouraging the three undesirable behaviors we had previously observed. The scenarios, defined in Python, are presented in Appendix A; and are visualized in Fig. 3, using an amalgamation of Statecharts and SBP graphical notation languages (Harel, 1987; Marron et al., 2018).

Scenario *avoid back-and-forth rotation* (Fig. 3(a)) seeks to prevent in-place, back-and-forth turns by the robot, to conserve time and energy.

Scenario *avoid turns larger than 180°* (Fig. 3(b)) seeks to prevent left turns in angles that are greater than 180°, to conserve time and energy (the right-turn case is symmetrical). A forward slash indicates an action that is performed when a transition is taken; square brackets denote guard conditions, and $k and $leftCounter are variables. Each turn rotates the robot by 30°, and so we set $k = 7$.

Scenario *avoid turning when clear* (Fig. 3(c)) seeks to force the agent to move towards the target when it is ahead, and there is a clear path to it. This is performed by blocking any turn actions when this situation occurs. Triggered events carry data, which can be referenced by guard conditions.

## 4    USING SCENARIOS IN DRL TRAINING

Even after defining constraints as an SB program, obtaining a differentiable function for the training process is not straightforward. We propose to use the following binary (indicator) function to this end:

$$c_k(s_t, a, s_{t+1}) = I(\text{the tuple } \langle s_t, a, s_{t+1} \rangle \text{ is a blocked state in the SB program, by the } k^{th} \text{ rule})$$

Intuitively, summing the values of the different $c_k$'s over the training episode yields the exact number of violations to the respective $k^{th}$ rule during the full trajectory; those are the values we aim to minimize; moreover, following the intuition of Roy et al. (2021), this value, if normalized over the number of steps, can be seen as a probability of having a violation. This value can be treated as a cost function, the corresponding objective function defined as follows: $J_{C_k} = \sum_N c(s_i, a_i, s_{i+1})$, for a trajectory of $N$ steps. This value is dependent on the action policy $a$ and is therefore differentiable on the parameters $\theta$ of the policy through the *policy gradient theorem*.

**Optimized Lagrangian-PPO.** In Section 2 we proposed to relax the Lagrangian constrained optimization problem into an unconstrained, *min-max* version thereof. Taking the gradient of Equation 2,

and some algebraic manipulation, we derive the following two simultaneous problems:

$$\nabla_\theta \mathcal{L}(\pi, \lambda) = \nabla_\theta (J_R(\pi) - \sum_K \lambda_k J_{C_k}(\pi)) \qquad \forall k, \quad \nabla_{\lambda_k} \mathcal{L}(\pi, \lambda) = -(J_{C_k}(\pi) - d_k) \quad (4)$$

In closed form, the Lagrangian dual problem would produce exact results. However, when applied using a numerical method like *gradient descent*, it has shown strong instability and the proclivity to optimize only the cost, limiting the exploration and resulting in a poorly-performing agent (Achiam et al., 2017). To overcome these problems, we introduce three key optimizations that proved crucial to obtaining the results we present in the next section.

1. *Reward Multiplier*: the standard update rule for the policy in a Lagrangian method is given in Equation 4. However, as mentioned above, it often fails to maximize the reward. To overcome this failure, we introduce a new parameter $\alpha$, which we term *reward multiplier*, such that $\alpha \geq \sum_K \lambda_k$. This parameter is used as a multiplier for the reward objective:

$$\nabla_\theta \mathcal{L}(\pi, \lambda) = \nabla_\theta (\alpha \cdot J_R(\pi) - \sum_K \lambda_k J_{C_k}(\pi)) \qquad (5)$$

2. *Lambda Bounds and Normalization*: Theoretically, the only constraint on the Lagrangian multipliers is that they are non-negative. However, when solving numerically, the value of $\lambda_k$ can increase quickly during the early stages of the training, causing the optimizer to focus primarily on the cost functions (Eq. 4), potentially not pushing the policy towards a high performance reward-wise. To overcome this, we introduced dynamic constraints on the multipliers (including the reward multiplier $\alpha$), such that $\sum_K \lambda_k + \alpha = 1$. In order to also enforce the previously mentioned upper bound for $\alpha$, we clipped the values of the multipliers such that $\sum_K \lambda_k \leq \frac{1}{2}$. Formally, we perform the following normalization over all the multipliers:

$$\forall k, \ \lambda_k = \frac{\tilde{\lambda}_k}{2(\sum_K \tilde{\lambda}_k)} \qquad \alpha = 1 - \sum_K \lambda_k \qquad (6)$$

3. *Algorithmic Implementation*: the primary objective of the previously introduced optimizations is to balance the learning between the reward and the constraints. To further stabilize the training, we introduce additional, minor improvements to the algorithm: (i) *lambda initialization:* we initialize all the Lagrangian multipliers with zero to guarantee a focus on the reward optimization during the early stages of the training (consequently, following Eq. 6, $\alpha = 1$); (ii) *lambda learning rate:* to guarantee a smoother update of the Lagrangian multipliers, we scale this parameter to 10% of the learning rate used for the policy update; and (iii) *delayed start:* we enable the update of the multipliers only when the success rate is above 60% during the last 100 episodes. Intuitively, this delays the optimization of the cost functions until a minimum performance threshold is reached.

## 5 EVALUATION

**Setup.** We performed training on a distributed cluster of HP EliteDesk machines, running at 3.00 GHz, with 32 GB RAM. We collected data from more than 100 seeds for each algorithm, reporting the mean and standard deviation for each learning curve, following the guidelines of Colas et al. (2019). For training purposes, we built a realistic simulator based on the Unity3D engine (Juliani et al., 2018). Next, we evaluated the performance of the trained models using a physical Robotis Turtlebot3 robot (Fig. 2) and confirmed that it behaved similarly to the behavior observed in our simulations.

**Results.** Fig. 4 depicts a comparison between policies trained with a standard end-to-end PPO (Schulman et al., 2017) (the baseline), and those trained using our constrained method with the injection of rules. In Figs. 4(a) and 4(d), we show results of policies trained with just *avoid back-and-forth rotation* added as a constraint. Fig. 4(a) shows that the success rate of the baseline stabilizes at around 87%, while the success rate of our improved policies stabilizes at around

95%. Fig. 4(d) then compares the frequency of undesired behavior occurrences between the baseline, at about 13 per episode, and our policies, where the frequency diminishes *almost completely*.

Next, for Fig. 4(b) we show results of policies trained with all three of our added rules; we note that the success rate for these policies stabilizes around 95%, compared to 87% for the baseline.

Finally, in Figs. 4(c), (e) and (f), we compare the frequency of the occurrence of undesired behaviors between the baseline and the policies trained with all rules active. Using the baseline, the frequency of the three behaviors is about 13, 3, and 17 per episode. The undesired behaviors are removed *almost completely* for the policies trained with our additional rules and method.

We note that the undesired behavior addressed by the rule *avoid turns larger than 180°* is quite rare in general; and so the statistics reported in Fig. 4(c) were collected over the final 100 episodes of training.

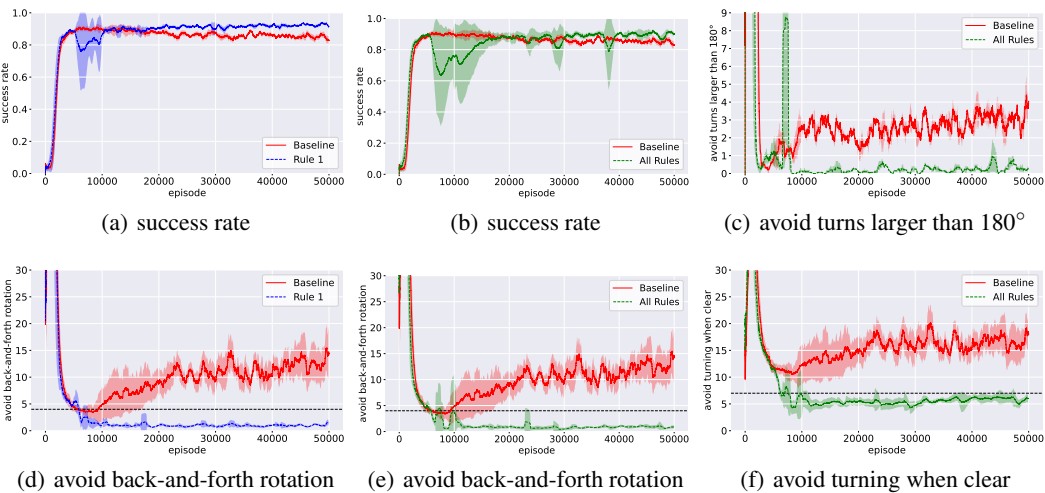

(a) success rate   (b) success rate   (c) avoid turns larger than 180°

(d) avoid back-and-forth rotation   (e) avoid back-and-forth rotation   (f) avoid turning when clear

Figure 4: A comparison between the baseline policies to policies trained using our approach. The black dotted line states the threshold ($d_k$) we considered for the $k^{th}$ rule.

The results clearly show that our method is able to train agents that respect the given constraints, without damaging the main training objective — the success rate. Moreover, it also highlights the scalability of our method, i.e., performing well when single or multiple rules are applied. Reviewing Fig 4(b), comparing the baseline's success rate with our method's success rate when all rules are applied together with all the optimizations presented in Section 4, shows a clear advantage. Excitingly, our approach even led to an improved success rate, suggesting that the contribution of expert knowledge can drive the training to better policies. This showcases the importance of enabling expert-knowledge contributions, compared to end-to-end approaches.

**Formal Verification and Safety Guarantees.** To further prove the effectiveness of our method, we show results of a DNN verification engine (Katz et al., 2019) assessing the reliability of our trained models. DNN verification is a sound and complete method for checking whether a DNN model displays unwanted behavior, over *all* possible inputs. We trained two batches of 60 models each: one batch of models trained by the "baseline" training algorithm, and one batch with models trained by our approach. The results indicate that while almost none of the baseline models upheld the properties (i.e., there was always at least one input for which they violated the properties), more than 80% of our improved models satisfied the properties for all possible inputs. A full summary of these results appears in Table 1 of Appendix B.

**Additional Environment.** To further validate our approach, we applied our algorithm to an additional environment beyond the one used in Section 5. In particular, we drastically increased the number of obstacles, which is one of the main challenges in mapless navigation. Fig. 5 shows the difference between the two training environments.

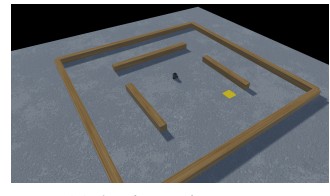 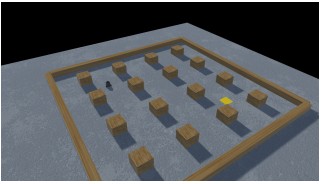

(a) basic environment (b) additional environment

Figure 5: A comparison between the basic training environment that we used for the main analysis (Sec. 5), and the additional environment used in this Section.

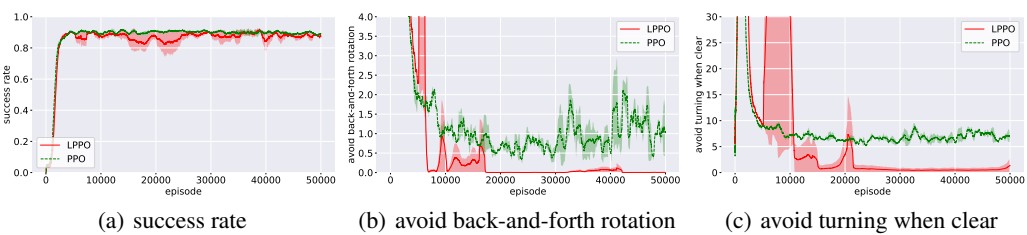

(a) success rate (b) avoid back-and-forth rotation (c) avoid turning when clear

Figure 6: Experiments in an additional, more complex, environment. The figure shows a comparison of our Lagrangian PPO (LPPO) and a baseline (PPO).

Fig. 6 shows the results of our experiments in a more complex environment. The results indicate that our approach scales very well, even when the number of obstacles is larger. Clearly, with respect to the results of Sec. 5, the algorithm requires more iterations to converge; however, the success rate satisfies the requirements, keeping the number of violations to the rule below the given threshold.

## 6 RELATED WORK

To the best of our knowledge, this is the first work that combines scenario-based programming into the training of a constrained deep reinforcement learning system. In Yerushalmi et al. (2022), the authors proposed integration between SBP and DRL using a reward-shaping approach that penalizes the agent when rules are violated, with an unconstrained optimization method. Our approach, based on constrained optimization, provides many advantages compared to the mentioned work, which results in high-performing agents and fewer rule violations. We provide an extensive comparison between the two approaches below.

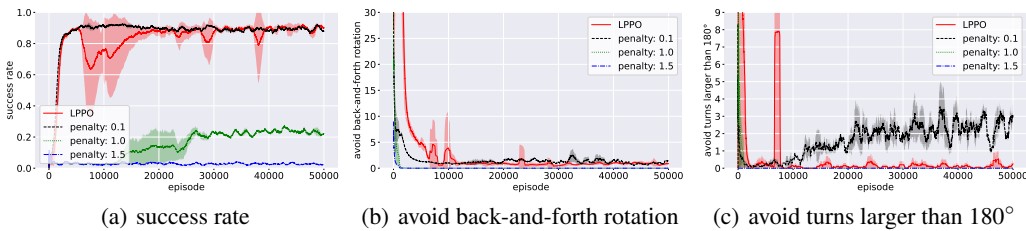

(a) success rate (b) avoid back-and-forth rotation (c) avoid turns larger than $180°$

Figure 7: The graphs compare results achieved by our approach, denoted by *LPPO*, with those achieved by (Yerushalmi et al., 2022), denoted by *penalty* and its value: fixed penalty of 0.1, 1.0, or 1.5: graph (a) measures the success rates with all three scenario-based rules. The results of using the *penalty* approach with a penalty value of 0.1 are practically the same as using our approach. However, using *penalty* approach with penalty values of 1.0 and 1.5 results in poor performance; graph (b) measures the frequency of violations to the *avoid back-and-forth rotation* rule. The results of using the *penalty* approach with penalty values of 0.1 and 1.0 are similar to ours. When using *penalty* approach with a penalty value of 1.5, the violations diminishes completely — however, also the performance, as mentioned above; graph (c) measures the frequency of violations to the *avoid turns larger than 180°* rule. The results of using the *penalty* approach with penalty values of 1.0 and 1.5 are practically the same as ours.

Fig. 7 compares the results of our approach and those of Yerushalmi et al. (2022). As shown in Fig. 7 using their approach, a low penalty value allows the agent to reach high-performance reward-wise but fails to minimize the cost (e.g., the number of rule violations). In contrast, a high penalty value reduces the agent's rule violations but fails to reach adequate performance in terms of the reward function. Our approach is shown here to reach similar performances as the best of (Yerushalmi et al., 2022), using a penalty value of $0.1$, and reducing the agent's rule violations as the best of it, using a penalty value of $1.0$ or $1.5$.

Our approach adopts a constraint-driven DRL framework that differentiates between optimizing the main reward and minimizing the costs. This differentiation presents significant advantages, including:

- Allows the setting of constraint thresholds independently for each rule/property and the handling of multiple such constraints in the same way, unlike methods such as (Yerushalmi et al., 2022) that only allow a global minimization to zero of the total cost.

- Separates reward maximization from cost minimization, simplifying the reward engineering task.

- Automatically balances the focus of the training, between the different cost elements and the reward, by learning the values of the different multipliers for each cost factor.

- Introduces novel numerical optimizations to the training phase, resulting in a more stable algorithm with a higher cumulative reward (as shown in Appendix C on a synthetic set of benchmarking environments).

In a recent work on constrained reinforcement learning (Roy et al., 2021), the authors advocate an optimized version of Lagrangian-PPO. They propose a different approach to balance the constraints and the return, based on the softmax activation function and without imposing bounds on the values for the multipliers. Moreover, their work focuses on game development, a different domain from our focus, which presents very different challenges, e.g., safety and efficiency are not considered crucial requirements. In addition, they do not encode constraints using a framework geared for this purpose, such as SBP.

**Limitations.** Our method suffers from various limitations. First, it does not completely guarantee that the resulting policies are safe. For example, as shown in Table 1 of Appendix B: even though the number of formally safe models is significant, it is not absolute. Second, the scalability of the method needs to be investigated. We showed in this work that the algorithm can easily handle one to three constraints, in addition to the main objective. We leave to future work the analysis of performance when the number of constraints increases further. Third, we noticed some performance deterioration after about 10,000 episodes shown in Fig.4 (a) and (b). We believe that the performance deterioration is related to the activation of the cost multipliers, especially as the performance was recovered afterward. We plan to investigate that further in the future.

## 7 Conclusion

This paper presents a novel and generic approach for incorporating subject-matter-expert knowledge directly into the DRL learning process, allowing to achieve user-defined safety properties and behavioral requirements. We show how to encode the desired behavior as constraints for the DRL algorithm and improve a state-of-the-art algorithm with various optimizations. Importantly, we define properties comprehensibly, leveraging scenario-based programming to encode them into the training loop. We apply our method to a real-world robotic problem, namely mapless navigation, and show that our method can produce policies that respect all the constraints without adversely affecting the main objective of the optimization. We further demonstrate the effectiveness of our method by providing formal guarantees, using DNN verification, about the safety of trained policies.

Moving forward, we plan to extend our work to different environments including navigation in more complex domains (e.g., air and water). Another key challenge for the future is to inject rules aiming to encode behaviours in a cooperative (or competitive) multi-agent environment.

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
