# OpenReview forum: "Constrained Reinforcement Learning for Safety-Critical Tasks via Scenario-Based Programming"
_ICLR.cc/2023/Conference — Submitted to ICLR 2023_

### Official Review · Reviewer_XYtz · 2022-10-24

**Confidence:** 4
**Correctness:** 4
**Technical Novelty And Significance:** 2
**Empirical Novelty And Significance:** 2
**Recommendation:** 3

**Clarity, Quality, Novelty And Reproducibility:**

The paper is clearly and well written. Novelty factor is low in my opinion (see comments above).

Some minor comments:

* eq. (2) max min should be reversed I think.
* I would add a reference for the Lagrange relaxation method per-se, e.g. the Bertsekas (Nonlinear programming) book.
* what is the difference of the scenario-based programming to the implementation of Finite State Machines?
* what happens when \eta and \beta are set differently?
* For (5) and (6) it would have been nice to discuss and compare to other Constrained RL approaches that follow the same Lagrangian relaxation approach. How do they deal with the identified problem of 'proclivity to optimize only the cost'?

**Strength And Weaknesses:**

* As a strength, I think that the idea to use scenario-based programming to inject rules easily into the optimization of safe-RL agents is a good idea, and the authors have picked a good application.

* However, I think that the contribution is quite minor. The approach can be subsumed, at least the author's formulation, within reward-shaping in general, as they cast the rules into constraints that are optimized together with the reward function. I think the introduced approach is not a new methodology but rather a successful reward shaping/modeling scenario for a particular experimental case.

* It is not clear why the introduced approach should be the winning one (compared to other e.g. reward shaping, approaches). An alternative formulation could have shaped the reward to include costs on the control actions, thereby effectively preventing some of these undesirable behaviour (if not all). Or the action space could have been defined appropriately (to avoid back and forth rotation). Or an MPC-like approach could have been followed to filter-out turns larger than 180 degrees for example. These are not discussed.

* As a related note, there are no comparisons to recent constrained/safe RL algorithms, only to a vanilla PPO approach. It is not motivated why only their approach can be used to train agents that satisfy the three rules they created.

**Summary Of The Paper:**

The authors introduce a new scenario-based programming approach where one can include rule-based constraints to safe-RL problems. By observing the suboptimality of trained PPO agents in a mapless navigation task, they come up with three rules that when implemented as additional constraints during the policy optimization, help remove the unwanted behaviors from the agent policy. The approach is compared to vanilla PPO that optimizes the safe-RL objective in the two experimental setups. The authors separately include a comparison to a similar scenario-based constrained RL approach, where they show improved behavior.

**Summary Of The Review:**

Overall I think the paper should be rejected: the contribution of the paper is quite minor, and the evaluations are not extensively done. Even if the experiments were significantly improved, I still think that the paper does not offer a significant method.

---

### Official Review · Reviewer_7ji2 · 2022-10-25

**Confidence:** 4
**Correctness:** 2
**Technical Novelty And Significance:** 2
**Empirical Novelty And Significance:** 2
**Recommendation:** 3

**Clarity, Quality, Novelty And Reproducibility:**

The writing is fairly readable but technical quality and depiction of related work could be improved. The paper appears somewhat original if rather niche.

**Strength And Weaknesses:**

Strengths:
- The problem is interesting as defining the reward function and constraints can be difficult.
- The authors test their approach on a real Turtlebot robot

Weaknesses:
- Scenario-based programming seems like yet another high-level state machine / visual programming language. As with all such syntactic representations for manually encoding rules, it is difficult to motivate why this representation is better than any other. Some kind of user study might have helped provide some evidence here.
- The proposed approach appears both fairly trivial and heuristic, (e.g. the reward multipliers, another hyperparameter).
- The RL baseline with the three failure scenarios seem suspiciously bad for such a simple environment. Imposing a constraint (make it a much harder problem) to keep the agent from turning back and fourth also seems like a complicated micromanagement of what might be some problem with the problem setup (state, reward). I am not convinced that the proposed approach is the best way to fix this.

The paper also contains some questionable statements like:
- "Specifically, the local nature of the problem renders learning a successful policy extremely challenging and hard to solve using classical algorithms (Pfeiffer et al., 2018)" - At a glance, I did not find any such strong claim in the referenced paper. It's a bit unclear what you mean by "local" here, but the mapless navigation problem used in this paper does not seem that difficult. A turtlebot can stop nearly instantly. You could just path to towards the goal with A* and replan on an incrementally built map. While there isn't any one algorithm that is *standard* for this problem (known goal, partially unknown environment), there is a lot of work on informative path planning and motion planning in unknown environments, if you need something more sophisticated than that.

- "Prior work has shown DRL approaches to be among the most successful for tackling this task, often outperforming hand-crafted
algorithms (Marchesini & Farinelli, 2020)." - That paper just appears to compare one RL-based formulation to another. It is not clear what you mean by "hand-crafted" algorithms, but I can't find anything about applicable planning algorithms.

- Eq.2: I think you mixed up the subscripts since you talk about maximizing reward?

**Summary Of The Paper:**

The paper proposes a way of defining the constraints in constrained RL via Scenario-based Programming. They demonstrate the approach on the problem of mapless navigation in simulation as well as on a real Turtlebot.

**Summary Of The Review:**

The problem of how to design reward/constraint functions is relevant, but the proposed approach is rather heuristic and not empirically well motivated (does it actually help people?), and the experiments are very simple.

---

### Official Review · Reviewer_ENWz · 2022-10-27

**Confidence:** 4
**Correctness:** 4
**Technical Novelty And Significance:** 3
**Empirical Novelty And Significance:** 3
**Recommendation:** 3

**Clarity, Quality, Novelty And Reproducibility:**

As outlined above, I find the paper very clear and of good quality. The novelty is (to the best of my knowledge) okay, but could use some more ‘sharpening’, as I will outline below. The results are reproducible and credible in my opinion. I have a few questions to the authors:

- Your reward setting is very non-sparse, via the distance function. I am wondering if your scenario approach could not help a lot in sparse environments, where RL agents usually have huge problems to converge. The programs could be designed in a way that the search space is reduced to more ‘relevant’ actions.
- Did you experiment with deactivating the ‘program controller’ at some point, to see if it is acting according to the program? This could be complementary to the verification step, and be an important feature to argue that the learning rate of the RL agent is good.
- In your implementation, why did you not use (or did you?) the mask() function of tensorflow.agents? This could be a simple way to block actions, and I think that some implicit, automatic, rescaling of rewards is performed.
- Please add some related work for the area of ‘shielding’ in RL, for instance:

Mohammed Alshiekh, Roderick Bloem, Rüdiger Ehlers, Bettina Könighofer, Scott Niekum, Ufuk Topcu:
Safe Reinforcement Learning via Shielding. AAAI 2018: 2669-2678

Nils Jansen, Bettina Könighofer, Sebastian Junges, Alex Serban, Roderick Bloem:
Safe Reinforcement Learning Using Probabilistic Shields. CONCUR 2020: 3:1-3:16



**Strength And Weaknesses:**

*Strengths*
- The paper is very well written and easily accessible.
- The concept is nice and effective.
- The type of domain knowledge via scenario-based programming is easily accessible for users, other than many other approaches that employ constrained RL.
- The evaluation is done on state-of-the-art environments.
- A formal verification step is additionally added to assess the quality of the trained network.

Weaknesses
- Some questions remain open regarding novelty and further evaluation.



**Summary Of The Paper:**

The authors of this paper introduce a new approach to safe (or constrained) reinforcement learning (RL) that leverages a concept called scenario-based programming. In essence, this programming paradigm is a means to inject domain knowledge to the RL loop, by running such a (set of) program(s) in parallel with the RL agent, as some form of external controller. A nice feature of these scenario-based programs is that they can capture a set of potential environments, and is therefore a good way to capture robustness of RL. The authors introduce these programs, describe in detail their embedding into RL agents, and introduce a dedicated ‘blocking’ mechanism that avoids actions not adhering to the program specification and rescales the reward/cost function accordingly, to avoid future violations. The evaluation is done on a turtlebot environment with unity as simulation engine.

--- after rebuttal ---
Since the authors did not reply to my questions, I recommend rejection for this submission
---


**Summary Of The Review:**

Good paper, related work/novelty can be better argued, some further experiments could strengthen the message.

---

### Decision · Program_Chairs · 2023-01-20

**Decision:**

Reject

**Justification For Why Not Higher Score:**

* The approach is quite heuristic
* It is not clear why the proposed framework should be better than alternative formulations
* The proposed approach seems to be subsumed by reward shaping
* There is no empirical comparison to existing constrained RL techniques

**Justification For Why Not Lower Score:**

NA

**Metareview: Summary, Strengths And Weaknesses:**

The paper proposes a constrained RL technique that leverages scenario-based programming.

Strength: the idea of scenario-based programming to inject domain knowledge is good

Weaknesses:
* The approach is quite heuristic
* It is not clear why the proposed framework should be better than alternative formulations
* The proposed approach seems to be subsumed by reward shaping
* There is no empirical comparison to existing constrained RL techniques

The above weaknesses are significant and therefore the paper is not ready for publication.